# Study on the Design Stage from a Dimensional and Energetic Point of View for a Marine Technical Water Generator Suitable for a Medium Size Container Ship

**Catalin Faitar [1,2,*] and Eugen Rusu [1]**

1   Department of Mechanical Engineering, Dunărea de Jos University of Galaţi, 47 Domnească Street, 800201 Galaţi, Romania
2   Faculty of Naval Electromechanics, Maritime University of Constanta, 104 Mircea cel Batran Street, 900663 Constanta, Romania
*   Correspondence: catalin.faitar@cmu-edu.eu

**Abstract:** The purpose of this study is to provide an overview of the development of the modern low-speed marine two-stroke diesel engine from the point of view of the technical water cooling plant, taking into account and starting from the market requirements for power and speed, with information and design options relevant to the entire shipping industry. Thus, through the ideas of this project, we analyze notions and relevant aspects of systems related to marine slow turning engines, including the basic thermodynamic structure of the technical water system designed for a marine engine. This study also presents the design criteria that define the size and design concept of the engine structure components, with a focus on the technical water cooling installation. The concepts for the main engine hot parts served by the technical water cooling installation play a vital role in the marine technical water generator. The choices the engine designer must make regarding basic auxiliary systems, such as fuel injection and exhaust valve actuation, are important factors to keep in mind when installing a technical water generator onboard a ship. The automation and control systems that govern the modern electronic engine, which drive the supply pump of the technical water cooling system can provide a simplified view of the engine development process. In order to point out the contributions of this study, it is important to focus on the calculations used to determine the main parameters of a technical water generator especially designed for a midsized container ship.

**Keywords:** technical; desalination; marine; seawater; operation; generator





## 1. Introduction

The requirements for the hot components of marine main engines are diverse and often conflict with each other. The temperatures of the components can reach values of 600 °C and even above, and these factors presuppose the correct choice of material and adequate cooling of these parts. Thermal stresses caused by temperature gradients must be considered in the analysis of low cycle behavior [1]. The components around the combustion space are bolted together with elastic pins, which are pre-tensioned so that there is no leakage and no dynamic sliding between the components.

Finding the adequate cooling principle is one of the main challenges for the piston designer. With the so-called jet cooling principle, which is applied to most marine main engines, it is possible to keep temperatures below the limit on the side of the combustion space as well as in the cooling holes. High temperatures (above expected values) for the piston tip would lead to material loss due to corrosive attack generated by the combustion gases. Excessive temperatures in the cooling space can lead to the accumulation of carbon deposits in solid form, with negative consequences for the cooling efficiency of the piston. For reliable engine operation, it is also necessary to maintain the temperatures around the piston rings within a certain predefined range.

It is important to keep in mind the basic knowledge regarding the cooling principles of the main components of a marine engine, especially when the main cooling fluid is salt water, which can damage sensitive parts built out of special materials and alloys. That is why technical water generators play such a crucial role in the operation of marine engines in an environment where fresh water is quite scarce. These systems have progressed in such a manner that they can sustain the optimal operation of 15.000 kW to 20.000 kW marine engines [2]. Technical water generators, also known as desalination systems/plants, have been used in the marine industry for quite some time now, but companies always try to improve their design and increase the degree of their usability [3].

Nevertheless, the authors would like to point out that the concept discussed in this paper represents a generalized configuration, one that could be applied to many similar systems that already exist in the technical water generating field as well as in the marine domain. Thus, the analysis and reinterpretations tend towards a generalized system customized for certain operating conditions onboard a commercial ship. That is why we have based our concept on the dimensioning of a technical water generator using established calculation principles to identify an optimal solution for a medium-sized container ship.

Similar concepts have also been addressed by other engineers, one patent relating to a marine water tank, and more particularly, to a marine water tank comprising an evaporation unit producing fresh water using the latent heat of steam [4]. This concept has been proposed a Korean scientist, via the application PCT/KR2013/010944.

The well-known Japanese marine systems company Mitsubishi, through application JP56164324A, developed a concept used to construct a water-making device for ships which uses the heat of heating fluid without the need for any evacuating means by providing a nozzle for ejecting supplied seawater directly toward the heat transfer tubes on the upper stage side of a heater. The concept provides an evacuating means such as orifice to a pipeline for supplying the seawater to said nozzle. This concept was used for a technical water generator installed on an oil tanker in 1995 [5].

On a different level, in 2017 a Russian scientist applied (application RU2017100751A) for a patent regarding an invention that relates to the field of machine building, in particular to seawater desalination installations (desalination units). The proposed desalination unit has at least two tanks to be filled with steam. Thermal compression of the steam in these steam tanks is performed by electric heaters [6]. Compressed steam is directed to the evaporator unit periodically from the first and second steam tanks. Removal of the remaining steam from the tanks is carried out in the low-pressure steam supply pipeline, using the heat of this steam to heat seawater. Supply, withdrawal and removal of steam from the steam tanks is regulated by the control system with the help of locking bodies.

More recently, Japanese scientists tried to invent and build a more complex system consisting of plants characterised by more than one engine delivering power external to the plant, the engines being driven by different fluids [7]. The engine cycles use thermally coupled combustion heat from one cycle to heat the fluid in another cycle with exhaust fluid of one cycle heating the fluid in another cycle. The potential of this concept has been presented in the application JP2018513870A.

## 2. Materials and Methods

In the first part of this study, special attention will be paid to the targeted compromise in terms of reliability, cost, manufacturability and maintainability of technical water generators, these being the aspects that challenge an engine designer (so-called licensor) to develop a competitive product to its customers, which are obviously shipowners, but also (licensed) engine manufacturers and shipyards. In addition, the tools and methods available for designing a technical water generator will be described in the context of the development process.

Furthermore, in subsequent sections, a detailed analysis of technical water generators is provided. Among the installations supporting the operation of a marine engine onboard a medium-sized container ships, we choose to examine the technical water cooling installa-

tion since it is not a very complex installation, but one that plays a crucial role in defining the life span of the main engine [8].

The main technical water sources onboard ships are briefly described in the third part of this study. It is important to underscore the fact that these systems have progressed, allowing ships to use additional spaces onboard (such as tanks) for other purposes than transporting fresh water.

Today, there are many types of technical water generators used in various industries. Depending on the main application domain, they vary in operational complexity. A brief classification is provided in the fourth section with a focus on the main processes used onboard ships to generate technical water.

In order to determine the amount of technical water required for a main engine onboard a medium-sized container ship, in the fifth section, we calculate the dimensioning of technical water generator installation, starting from the premise that all the essential data relating to engine power, vessel size, technical water requirements and other relevant data are known. The container ship used is part of the Panamax category.

Assuming that the dimensioning calculation meets our objectives, we establish an optimal model of a technical water generator which meets the requirements of the ship's main engine and which ensures its operation at the nominal parameters during a voyage of normal duration. Afterwards, taking into account the dimensional and energy constraints onboard the ship, in the final part of the study, we present a method to optimize the operation of the technical water generator. In this way, a series of technologies suitable for this type of system are identified after studying similar models already tested in the marine market.

Seawater contains a high percentage of salts, which makes it unusable for the sanitary installations on ships or as a heating agent in the power installations. Overall, seawater salinity is defined as the weight of all ions contained in a liter of water and is expressed in g/L [9]. Salinity depends on the geographical area; its average value can be considered 33–37 g/L.

Obtaining technical water or fresh water from seawater is carried out in complex installations based on various functional principles. The thermal installations for desalination by distillation and by freezing have been known for a long time. For the purpose of This study examines the operation of common commercial marine technical water generator units which are mainly based on commercial thermal units, most of which can be found on cargo ships and in other marine applications (e.g., oil rigs and passenger ships).

The desalination mechanism can be easily explained if we use a simplified representation of the composition of seawater that includes only water molecules, $Na^+$ and $Cl^-$ ions. Water molecules, considered as dipoles, surround the ions, forming a layer of molecules attached to the ions by electrostatic forces. The ensemble formed by ions and attached molecules is much larger than the free molecules, and for desalination the two parts must be separated. For technical water to be usable, the salt content must be below 10 p.p.m. (parts per million). When the salt concentration is 18–19 p.p.m. the water can be used for cooling, and if this concentration is even lower (1...3) p.p.m., the water can be used to feed the boilers. Desalination of seawater, in order to use it as technical or drinking water, can be carried out industrially through the following different processes [3,10,11]:

- Distillation;
- Freezing, which can be

  - Direct freezing under vacuum, through vapor absorption;
  - Direct freezing under vacuum, by vapor compression;
  - Indirect freezing with secondary refrigerant (classical method);
  - Indirect freezing with n-butane type secondary refrigerant.

- Reverse osmosis [12].

The conceptual process described in this study is designed to produce boiler feed water with a salinity of <5 p.p.m. The first methods those listed above can be achieved by

applying the phase-changing process, which involves high energy consumption and are already considered classics. All processes related to desalination are based on the physical and sometimes chemical properties of both pure water and aqueous solutions (electrolytes), as seawater is considered to be an electrolytic solution.

Taking into account the low energy consumption and the easy maintenance of a technical water generator (or the marine plant) that operates on the basis of reverse osmosis, the quality of the water thus obtained will be reviewed [8,13].

Reverse osmosis is essentially a special process of ultrafiltration of a saline solution using semipermeable synthetic membranes with an appropriate chemical constitution instead of a filter with a porous structure. If appropriate pressure is applied to the saline water in a vessel, which is in contact with such a membrane, the pure water will pass through it, the hydrated ions being retained. Reverse osmosis has practical applications in technical water generators used onboard ships [14].

The principle of reverse osmosis can be better understood if we know the phenomenon of osmosis, which is explained as follows: when two aqueous solutions of unequal concentrations (one concentrated and the other diluted) or a solution and pure water are separated by a membrane of a certain type, water molecules from pure water, or the dilute solution, pass through the membrane into the concentrated solution [15,16].

The principle behind the operation of the electrodialysis process of a marine technical water generator is shown in Figure 1:

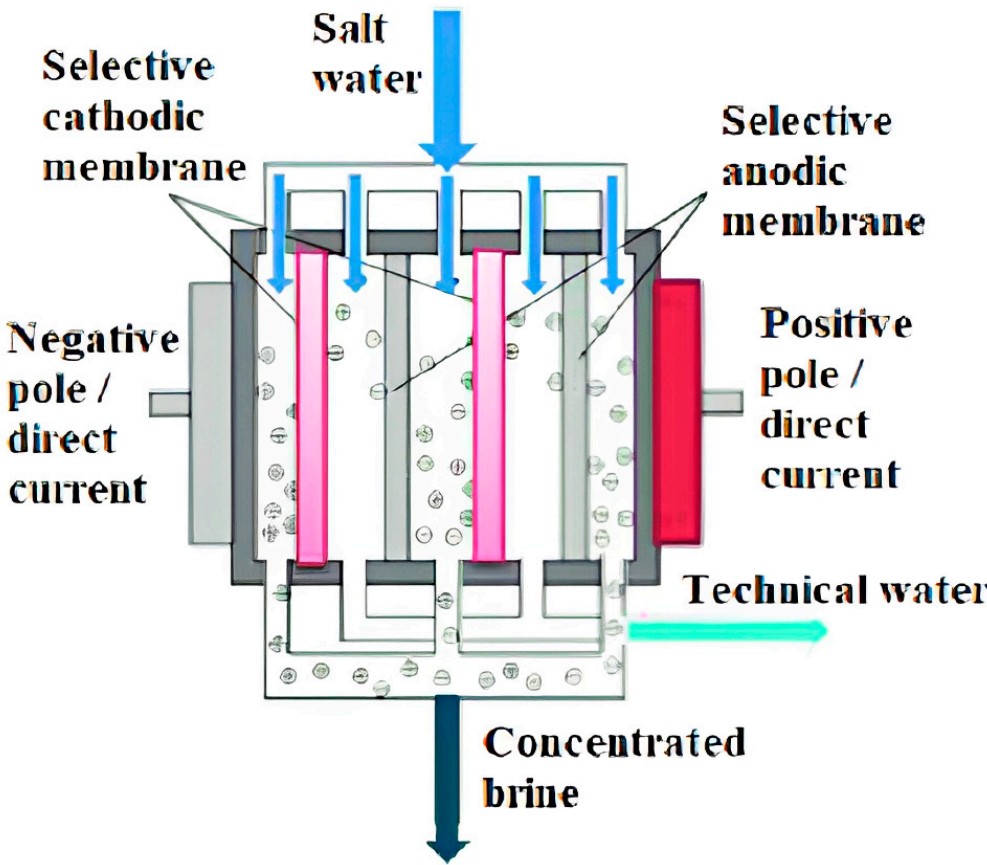

**Figure 1.** The principle of operation of the electrodialysis process of a marine technical water generator [drawn by the authors].

For many years, increasing interest has been shown in the conversion of seawater into drinking or technical water. To meet the growing need for technical water both onshore and onboard ships, much research has been conducted to find effective methods of removing salt from seawater [17]. Maritime ships have a great need for drinking water, as well as

for technical water for washing, for cooling the engines and for many other activities. In the past, ships carried large quantities of fresh water in barrels or specially built tanks because there was no source of fresh water when the ship was at sea except when it rained. Nowadays, despite the fact that fresh water is produced onboard, ships still have to load tons of fresh water. Although technological methods of converting seawater to fresh and technical water exist, they cannot meet the needs onboard ships [18].

The potential lack of technical water is also due to the fact that it is very difficult to control and limit consumption, and it is also impossible to avoid the consumption of this vital element. Research has been conducted and several processes have been developed to obtain technical water from seawater. These processes are distillation, electrodialysis, reverse osmosis, direct freezing evaporation and vapor compression distillation [19,20].

The term reverse osmosis derives from a well-known process inspired by nature itself. This means that if we put a concentrated and a diluted solution in two containers separated by a very fine membrane, the system will try to balance itself in a natural manner due to the fact that the diluted solution will pass through the membrane and mix with the concentrated solution until the same concentration is reached in both containers. The height of the liquid column in the container which holds the concentrated solution will increase until the column becomes too high, and the pressure exerted by it will stop the flow of the diluted solution. The balance point of the water column is called the osmotic equilibrium. If a force is applied to this column, the direction of flow of the solution through the membranes can be changed. This is actually reverse osmosis [21].

In Figure 2 below, the process of the reverse osmosis installation is shown. Seawater is drawn in by a low-pressure pump which in turn feeds a high-pressure pump (with a nominal pressure set over 60 bar). The high-pressure pump then circulates the seawater into the membrane, and due to this process combined with high pressure and the density of the seawater, the brine will be separated from the potable water, so at the other end of the membrane, two circuits will be generated, one of fresh water and one of brine (that will be pumped overboard). As seen in Figure 2, the installation can also be equipped with sand filters and micron filters used to separate impurities from the water, needed to protect the membranes [22].

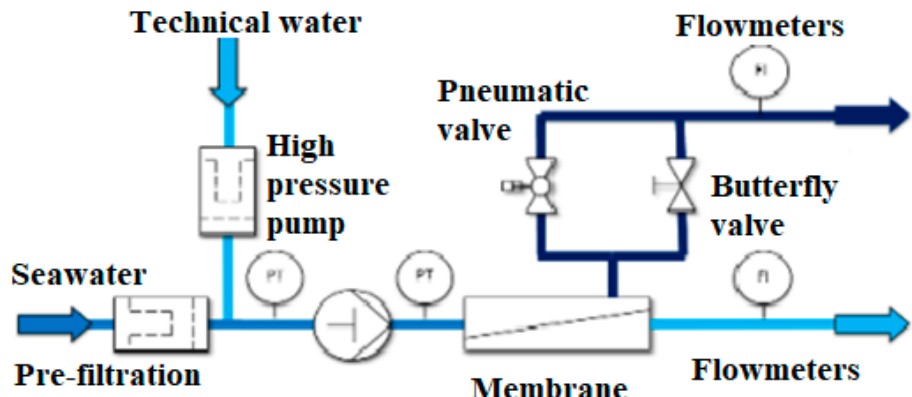

**Figure 2.** Diagram and model of a marine osmosis installation [drawn by the authors].

## 3. Results

As a general rule, water installations and sanitary installations (drinkable or technical), installed onboard ships are similar. The only element that differs is the type of installation chosen. The only types of ships for which stricter conditions are imposed are passenger ships, which must meet a series of very demanding quality parameters, both in terms of drinking water and ballast and wastewater. This can be an issue if the system is being installed onboard a smaller ship because in order to keep potable water onboard in storage, the ship must be equipped with at least two tanks [23].

Onboard the ship, the water installation has three well-defined roles, namely: supplying drinking water to the crew and passengers, supply technical water to power the power units and cabins, and supply seawater to flush the toilets and use in washing machines. Corresponding to the three roles, the water supply installation onboard the ship will be composed of three installations that operate according to the same principle and have the same common elements, with a few differences: the technical water installation can use structural tanks not covered on the inside as storage tanks, and it includes a boiler for heating domestic hot water; the seawater installation takes water directly from the seawater main system in a similar concept as that drawn in Figure 3.

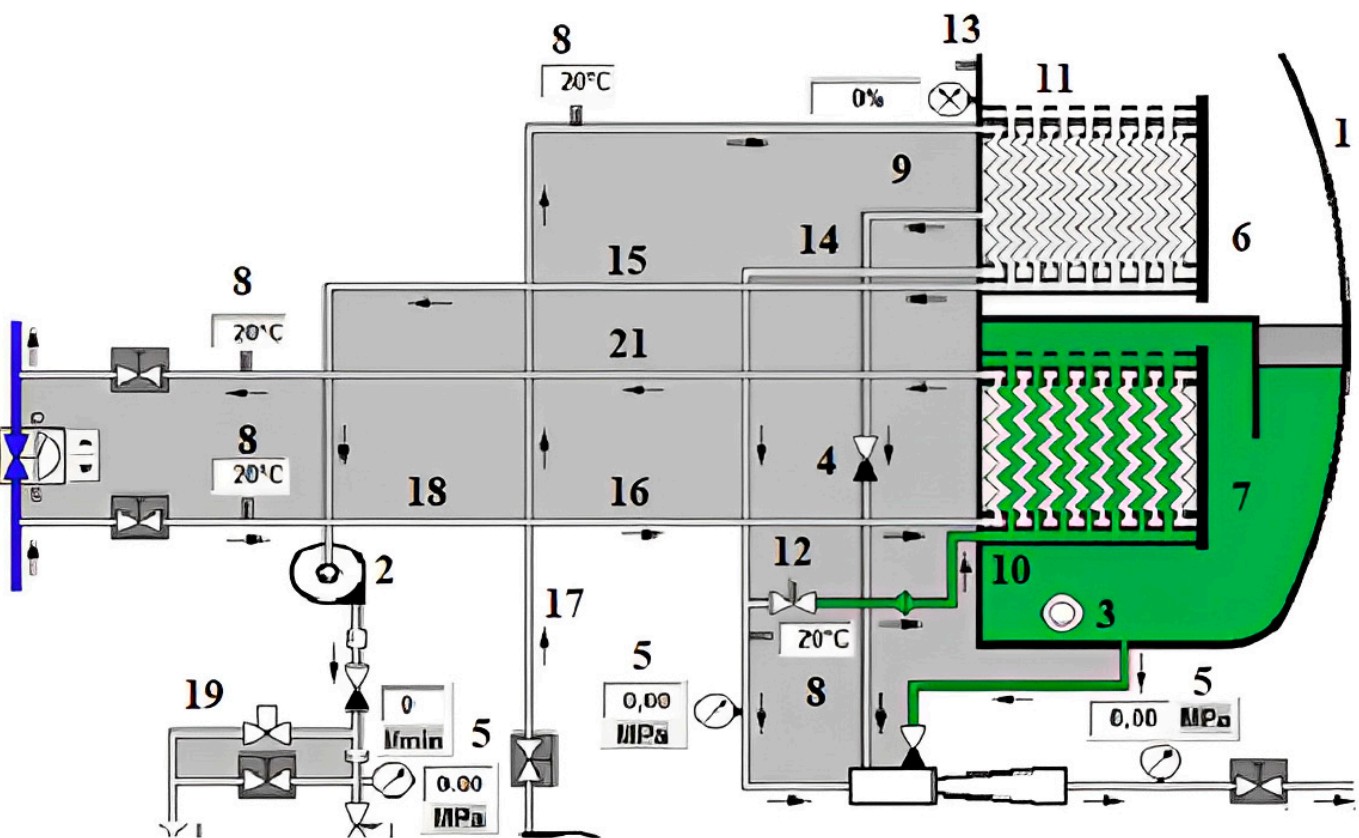

**Figure 3.** Typical diagram for the water installation onboard a ship [drawn by the authors].

The general technical water system contains the following elements in its composition:

1. Technical water storage tank;
2. Water supply pumps;
3. Water pump for water (potable, technical, sea);
4. Retaining flap;
5. Pressure relay;
6. Technical water treatment module;
7. Technical water module cooling mode;
8. Temperature transducer;
9. Technical water consumption equalization tank;
10. Circulation pump;
11. Boiler (with steam);
12. Valve controlled by transducer for steam input;
13. Tank ventilation equipment;
14. Supply piping coming from the technical water generator;
15. The distribution network for cold technical water;

16. The distribution network for hot technical water;
17. Main seawater supply;
18. Distribution network for seawater;
19. The main piping line of the compressed air installation;
20. Hydrophore safety valve;
21. The distribution network for technical water [24].

Regarding technical water, the installation and operation are similar with the difference that cold technical water is distributed to consumers through parallel pipes and hot technical water through adjacent pipes. This principle has already been pointed out in Figure 2, illustrating the process flow for a similar RO (reverse osmosis) unit. The water is heated in the boiler, which can be with steam, hot water, burner or electric. Regardless of the type of boiler, the installation includes: a circulation pump, an equalization tank and a temperature transducer that controls the start/stop of the pump and the control of the steam valve in order to obtain the preset temperature [25].

The flow rate of technical water pumps is determined as follows [25]:

$$Q_p = \sum_i (n_i \alpha_i q_i) \tag{1}$$

where n is the number of consumers of the same kind, $\alpha$ is the simultaneity coefficient of the operation of consumers of the same type, and q is the consumption norm in [L/s] specific to each type of consumer.

The values of the coefficient and the consumption norm depending on the type of consumer are as it follows:

- For category 1 consumers: $\alpha = 0.3$ and $q = 0.07$ L/s;
- For category 2 consumers: $\alpha = 0.2$ and $q = 0.2$ L/s;
- For category 3 consumers: $\alpha = 0.4$ and $q = 0.15$ L/s;
- For category 4 consumers: $\alpha = 0.5$ and $q = 0.3$ L/s;
- For category 5 consumers: $\alpha = 0.25$ and $q = 0.2$ L/s [25].

Therefore, according to the above values, we will consider:

$$\alpha_1 = 0.3; \alpha_2 = 0.2; \alpha_3 = 0.4; \alpha_4 = 0.5; \alpha_5 = 0.25$$
$$q_1 = 0.07; q_2 = 0.2; q_3 = 0.15; q_4 = 0.3; q_5 = 0.2$$

The number of consumers are set as:

- For category 1 consumers: $n_1 = 55$;
- For category 2 consumers: $n_2 = 48$;
- For category 3 consumers: $n_3 = 5$;
- For category 4 consumers: $n_4 = 4$;
- For category 1 consumers: $n_5 = 4$.

The required flow of the installation will be (Equation (3) [25] and Equation (5) [25]):

$$Q_p = n_1 \alpha_1 q_1 + n_2 \alpha_2 q_2 + n_3 \alpha_3 q_3 + n_4 \alpha_4 q_4 + n_5 \alpha_5 q_5 \tag{2}$$

$$Q_p = 4.175 \frac{1}{s} \tag{3}$$

$$Q'_p = Q_p 3.6 \tag{4}$$

$$Q'_p = 15.03 \frac{m^3}{h} \tag{5}$$

The calculation of the working volume of the hydrophore is performed as follows: if i = 7…9 represents the number of starts per hour, we consider i = 8 for the calculation. Its volume will be [25]:

$$V = \frac{Q'_p}{i} \rightarrow V = 1.879 \ m^3 \tag{6}$$

Calculation of hydraulic losses [7]:

$$h = \lambda \frac{I_c}{d} \rho \frac{v^2}{2} \tag{7}$$

where $v_{rec} = 1 \dots 1{,}2 \, \frac{m}{s}$ is the recommended speed in the pipes $v = v_{rec}$, thus $v = 1 \frac{m}{s}$.

Pipe diameter is calculated as [26]:

$$d' = \sqrt{\frac{4 Q_p}{\pi v_{rec}}} \, [m] \rightarrow d' = 0.0729 \, m \tag{8}$$

The diameter is standardized, so the number of centimeters is [26]:

$$nr_{cent} = \frac{d_{mm}}{25.4} \rightarrow nr_{inch} = 2.87 \, cm \tag{9}$$

Thus, $d_m = d \cdot 10^{-3}$.

The kinematic viscosity is [26]:

$$v = 1.287 \cdot 10^{-6} \frac{m^3}{s} \text{—at } 12 \, °C \tag{10}$$

The Reynolds number will be [26]:

$$R_e = \frac{v d_m}{v} = 5.921 \cdot 10^4 \tag{11}$$

Welded, newly galvanized steel pipes with the height of roughness for this type of piping is considered as [26]:

$$K = 0.1 \dots 0.2 \, mm \text{ and } \varepsilon = \frac{k}{d_{mm}} = 2.057 \cdot 10^{-3} \tag{12}$$

Altsul's criterion is applied, and the next values will be calculated [26]:

$$R_{e1} = \frac{10}{\varepsilon} = 4.861 \cdot 10^3 \text{ and } R_{e2} = \frac{500}{\varepsilon} = 2.43 \cdot 10^5 \tag{13}$$

The condition $R_{e1} < R_e < R_{e2}$ is met. It follows that the pipe is hydraulic semi-rough, and we will calculate the coefficient of hydrodynamic friction as follows [26]:

$$\lambda = 0.11 \left( \varepsilon + \frac{68}{R_e} \right)^{0.25} = 0.026 \tag{14}$$

The value of the calculation length $l_c$ and the equivalent length $l_e$ is calculated according to the technical specification for the tank type ship. For the calculated diameter of the piping, we will consider the following equivalent lengths:

- $l_v = 25$—for the check valve;
- $l_c = 5.2$—for a bent;
- $l_t = 5.2$—for normal tee conjunction;
- $l_{ct} = 14$—for sip;
- $l_{CT} = 0.55$—for fully open valve [6].

The equivalent length is [26]:

$$l'_e = 2 l_v + 7 l_c + 2 l_t + 1 l_{ct} + 5 l_{CT} = 113.55 \, m \tag{15}$$

The calculation length is [26]:

$$l_c = (10 + 2 + 1 + 0.7) \cdot 12 = 164.4 \, m \tag{16}$$

The pump load given by linear and local losses will be [26]:

$$h = (\lambda \frac{l_c}{d_m} + \lambda \frac{l'_e}{d_m})\rho \frac{v^2}{2} = 4.774 \cdot 10^4 \text{ Pa} \tag{17}$$

The total load of the installation is given by the sum of the losses on the piping and the losses given by the height at which it discharges [26]:

$$H_i = \rho g z + h = 1000 \cdot 9.81 \cdot 16.4 + 4.774 \cdot 10^4 = 2.086 \cdot 10^5 \text{ Pa} \tag{18}$$

The minimum pressure in the hydrophore is [26]:

$$p_1 = H_i = 2.086 \cdot 10^5 \text{ Pa} \tag{19}$$

The maximum pressure in the hydrophore is [26]:

$$p_2 = \alpha p_1 = 1.6 \cdot 2.086 \cdot 10^5 = 3.755 \cdot 10^5 \text{ Pa} \tag{20}$$

The pump load is equal to the maximum pressure in the tank [26]:

$$H_{pump} = P_2 = 3.755 \cdot 10^5 \text{ Pa} \tag{21}$$

$$H'_{pump} = \frac{H_{pump}}{\rho \cdot g} = \frac{3.755 \cdot 10^5}{1000 \cdot 9.81} = 29.773 \text{ m water column} \tag{22}$$

The flow rate of the feed pump is [26]:

$$Q_{pump} = \frac{Q'_p}{0.85} = 17.682 \frac{m^3}{h} \tag{23}$$

## 4. Discussions

According with the calculation at the end of the previous section, the main technical characteristics of the water installation onboard the container ship were determined. The data obtained through this calculation is essential to establish the component elements within the system framework. Further, on the basis of the same calculation, the technical water generator that serves this installation and for which the operation optimization study will be carried out will be established.

The chosen concept of a technical water generator has an optimized operation, being designed under the "start-and-forget" operational concept ("start it and forget it") and being easily arranged in compartments with machines that do not require constant human supervision. It can also be easily integrated into fully automated operations. This technical water generator is specially designed for operation onboard ships or onboard oil platforms. [27] The operating principle of the desalinator is based on a type of technology unique to this market that uses 3-in-1 type plates that allow desalination to take place in a single pack of plates without the need to install an external plate [27]. Evaporation, separation and condensation processes will all occur within this single plate pack, as represented in Figure 4. The construction of the plates is optimized from the start by the nature of the metal used for their construction, a titanium alloy.

Depending on the needs and configuration of the ship, it can be customized as any type of project that involves the integration of a technical water generator with a wide range of configurations [28].

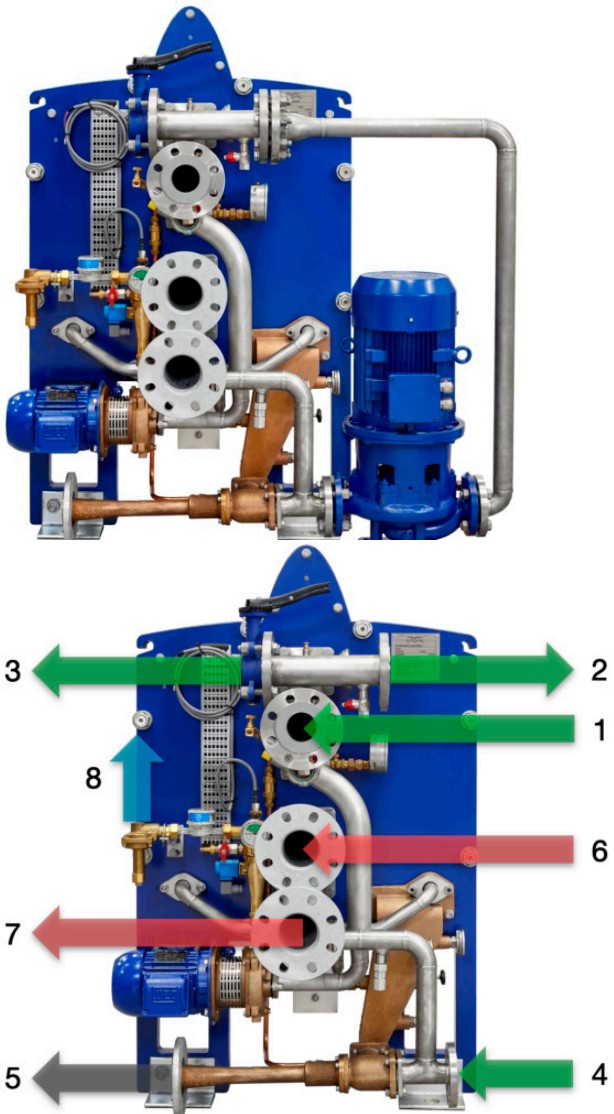

**Figure 4.** The technical water generator with the main component elements [drawn by the authors].

The figure above shows the technical water generator onboard the reference ship with its main component elements as follows:

○      1—Suction of seawater/cooling water;
○      2—Ejector discharge/feed water;
○      3—Discharge of seawater/cooling water;
○      4—Ejector suction/feed water;
○      5—Brine discharge;
○      6—Suction heating medium (hot water or steam);
○      7—Discharge of heating medium (hot water or steam);
○      8—Freshwater discharge.

The equipment onboard the midsize container ship has the following additional equipment intended to optimize its exploitation:

-      Supply or ejector pipes;
-      Technical water discharge bypass system;
-      "Built-in" control panel;
-      "Built-in" water treatment equipment. This is to support its three main roles;
-      Special valves for manometers;
-      Pressure gauges on the ejector, water pump and hot water pump;

- Filling line with dosing system and counter-flanges;
- Protective cover for plates;
- Distance indicators on the fixing bolts;
- Hot water pump;
- Hot water loop for connecting an additional heat source;
- Self-cleaning unit (CIP);
- Spare parts kits;
- UV sterilization unit of the treated water;
- pH adjustment filter;
- Chlorination unit;
- Dechlorination unit;
- Sterilizer with silver ions;
- Additional mode for measuring the purity of technical water [29].

### 4.1. Optimizing the Operation through the Installation Mode

The technical water generator is very easy to install, as Figure 5 demonstrates. Since the time required for repairs or maintenance work is limited (and this is a positive aspect, taking into account the limitations onboard ships), the installation is very compact. The heating medium in which the still operates is hot water, such as water coming from the main engine, but it can also be steam. The cooling liquid of the condenser is sucked directly into the cooling system of the main engine, to which it is also returned after passing through the distiller system. The feed water is sent by means of a pump or an ejector, this being necessary for the evaporation process, as well as the water that must reach the ejector of brine combined with air. This water can be drawn from the condenser discharge, or it can come from an alternative source.

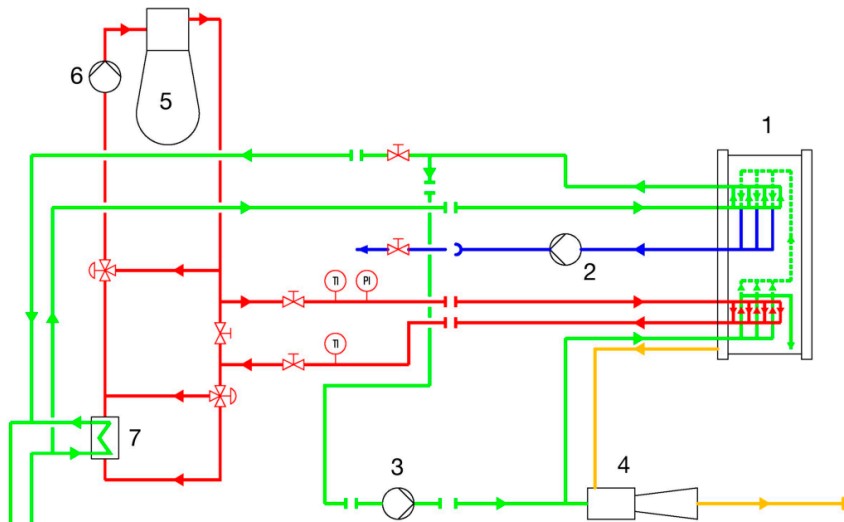

**Figure 5.** The arrangement of the AQUA Blue S-type technical water generator in the framework of the cooling water installation [drawn by the authors].

The technical water generated is pumped to a storage tank. The control panel provides electrical power to the feed pump, to the ejector, to the water pump and to the metering pump and also provides the control voltage to the hallometer and the drain valve. The components of the diagram above are as follows:

○ 1—Technical water generator;
○ 2—Technical water pump;
○ 3—Water supply pump;
○ 4—Brine/air ejector;
○ 5—The main engine of the ship;

○   6—Cooling water system;
○   7—The central cooler of the main engine.

### 4.2. Optimization of Electricity and Water Consumption

The technical water generator has an optimized technology that allows it to convert a higher percentage of technical water from salt water. More than that, it cuts the amount of salt water needed for cooling in half. This means that a halved amount of electricity is consumed for pumping, representing half of the electricity consumption dedicated to the operation of the entire system. For cooling, it is possible to opt for the still to be cooled directly by the open cooling circuit of the main engine, and this would further reduce electricity consumption [30]. This difference is highlighted in the diagram in Figure 6.

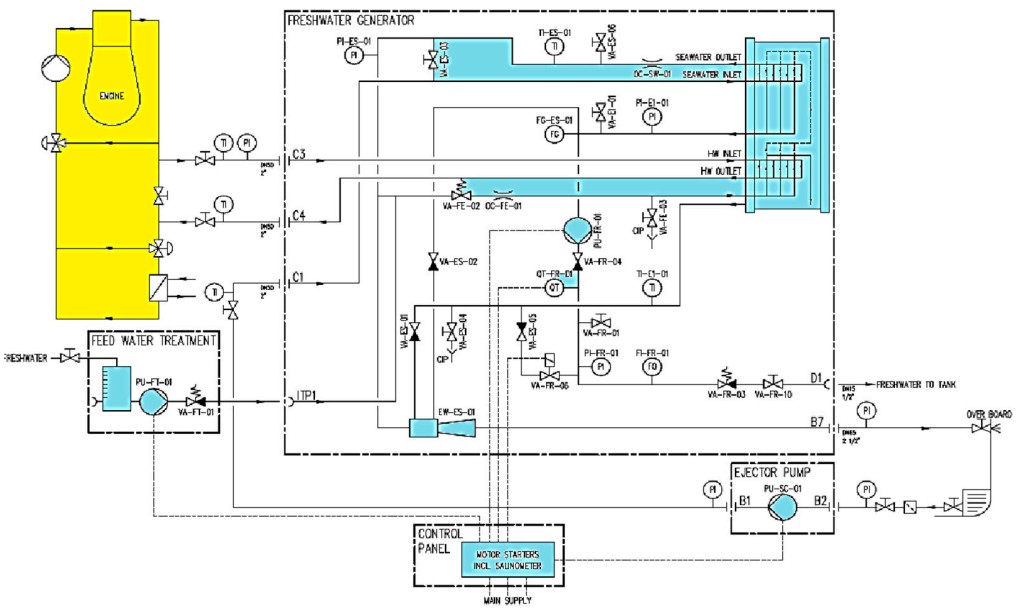

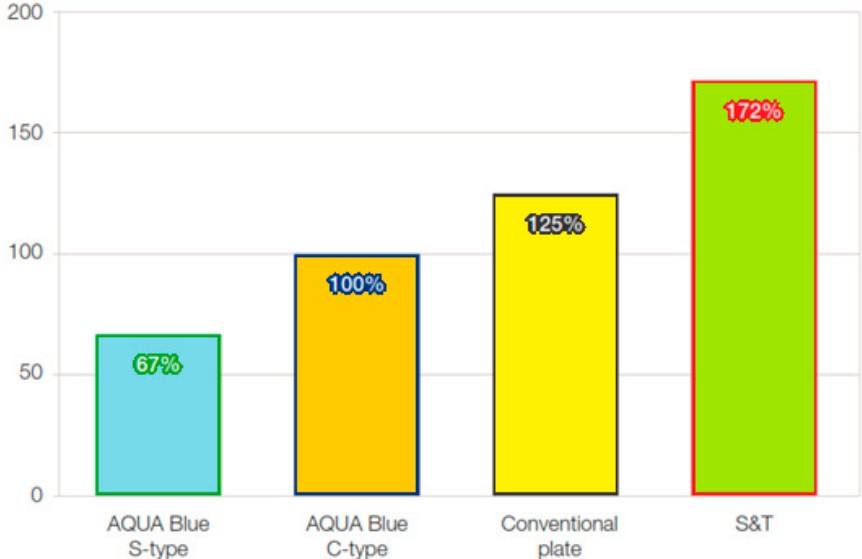

**Figure 6.** Figure of electricity consumption compared to other types of technical water generators—with aqua blue C type diagram shown above [18] [drawn by the authors].

### 4.3. Optimization through the Use of Membranes with Special Construction

As mentioned, the system is composed of titanium plates with a 3-in-1 design, the main differences between these plates being marked in Figure 7. They are provided with

two types of gaskets, and the whole configuration allows evaporation, separation and condensation in a single package uncovered by the plates. At the same time, this specific configuration enhances the evaporation process and allows the conversion of a larger amount of salt water.

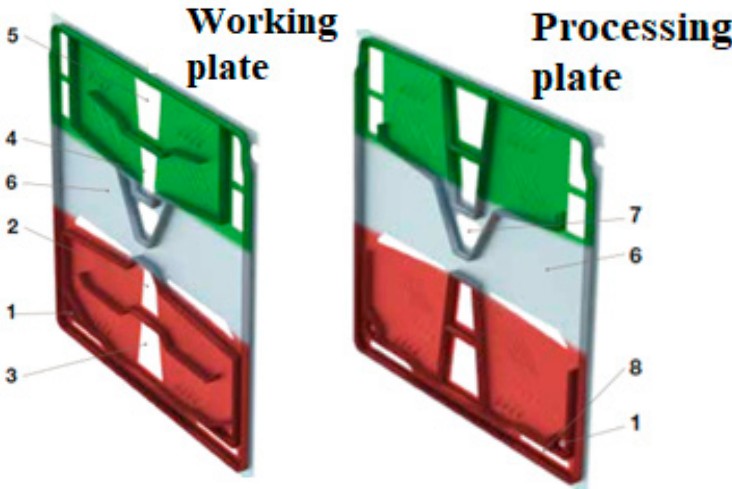

**Figure 7.** Model of titanium plates used in the configuration of the freshwater generator [drawn by the authors].

In the figure above, the colors associated with the areas of the titanium plates denote the following processes:

○　Green—condensation;
○　Gray—separation;
○　Red—evaporation.

The constructive elements of the plates, as they are numbered in the figure above, are:

○　1—Saltwater supply;
○　2—Input of the heating medium;
○　3—Evacuation of the heating environment;
○　4—Suction of sea/cooling water;
○　5—Discharge of sea/cooling water;
○　6—Evaporated steam;
○　7—Evacuation of technical water;
○　8—Brine evacuation [31]

The salt water passes through the lower part of the plate pack, where it is evaporated at a temperature of 40–60 °C in a vacuum process of 85–95% [32]. As the steam increases its pressure and density between the plates, it passes through the separation zone, which locks the brine and makes it fall into the pool at the base of the technical water generator. Thus, only technical water vapor reaches the upper part, which is cooled and condensed into fresh water. The result of the whole process is the generation of good quality water with a maximum salinity of 5 ppm [33]. Due to the optimized flow distribution along the plates, any form of calcareous deposit is avoided.

*4.4. Optimizing by Ensuring Correct Exploitation and Compliance with the Installation Maintenance Schedule*

Although the technical water generator does not require a high level of maintenance, its operation depends in many ways on the way the installation it serves is maintained. For these reasons, the maintenance of these types of devices is closely related to the maintenance technical water installation onboard the ship [34]. In general, the maintenance and upkeep of the installations onboard ships must focus on the following directions:

-　General visual check;

- Checking pump operating parameters;
- Checking the degree of corrosion;
- Check/maneuver valves and taps;
- Checking connections;
- Checking the fixing supports of the component parts (especially in the case of the hydrophore);
- Checking the water reserve and level;
- Verification of the expansion vessel;
- Checking of measuring and control devices;
- Check dashboards;
- Periodic function tests, especially when the installation is not turned on for long periods of time;
- AMC metrological verification (measurement and control devices) of the installation;
- Checking the external/internal water tank and control and signaling devices, cleaning, periodic repainting;
- Ensuring safety lighting in the spaces that serve such installations by checking the electrical lighting installations and lighting fixtures [35].

The control and checks of the installation have a permanent character, being part of the current follow-up regarding the technical condition of the construction, which, correlated with the maintenance and repair activity, has as its objective the maintenance of the installation at the designed parameters. The control and verification of the installation is performed by the operating personnel on the basis of a program. The program will include provisions regarding the entire installation, the categories of installation elements and functional operations, recorded in the operating instructions of the installation.

## 5. Conclusions

Desalination of ocean water requires a huge amount of energy and of course produces greenhouse gases. Moreover, desalination plants endanger marine ecosystems. In a report dedicated to ocean water desalination plants, representatives of the environmental organization WWF were quite concerned about this issue. Seawater desalination is far from the ideal solution. This technology is a potential threat to the environment and will only make climate change worse [36]. The recourse to these new technologies, which are increasingly accessible, will not remain without consequences for the environment.

Technical water systems or plants can also have a negative impact on coastal areas, amplifying the destruction of marine ecosystems specific to these areas. In addition, the risk of disturbing the balance of these wetlands and the purification and protection functions against catastrophes increases greatly.

Specialists in marine pollution and ecotoxicology have also sounded the alarm about the dire consequences that could occur after the commissioning of the desalination unit onboard the ships [37]. The process of desalination and drinking water production plays a critical role when it comes to navigation and making long-distance and long-duration journeys.

At the same time, the desalination process plays an important role in many other fields, totally different from maritime transport. Many countries in arid areas use seawater as a source of drinking water for their coastal cities, subjecting it to expensive desalination processes. Obviously, the oldest and largest desalination plant is nature itself through evaporation from the seas. More than 30 water desalination processes are known, including condensation, freezing, extraction, electrodialysis, reverse osmosis, ion exchangers, etc.

**Author Contributions:** Conceptualization, C.F. and E.R.; methodology, E.R.; software, Microsoft Office Word; validation, C.F. and E.R.; formal analysis, E.R.; investigation, E.R.; resources, C.F.; data curation, C.F.; writing—original draft preparation, C.F.; writing—review and editing, C.F. and E.R.; visualization, C.F. and E.R.; supervision, E.R.; project administration, E.R.; funding acquisition, C.F. All authors have read and agreed to the published version of the manuscript.

**Funding:** This research was funded by PROINVENT – POCU Programme grant number PN-III-P4-ID-PCE-2020-0008 And The APC was funded by Romanian Executive Agency for Higher Education, Research, Development and Innovation Funding—UEFISCDI.

**Data Availability Statement:** Further data on author work can be studied by accessing the following links, for additional information: https://iopscience.iop.org/article/10.1088/1757-899X/1182/1/012024/meta, https://iopscience.iop.org/article/10.1088/1757-899X/1182/1/012023/meta, https://www.gup.ugal.ro/ugaljournals/index.php/mtd/article/view/4028, https://www.gup.ugal.ro/ugaljournals/index.php/mtd/article/view/4028.

**Acknowledgments:** The work of Faitar Catalin was supported by the project "PROINVENT", Contract no. 62487/03.06.2022—POCU/993/6/13—Code 153299, financed by The Human Capital Operational Programme 2014–2020 (POCU), Romania, the work of E.R. was carried out in the framework of the research project DREAM Dynamics of the Resources and technological Advance in harvesting Marine renewable energy), supported by the Romanian Executive Agency for Higher Education, Research, Development and Innovation Funding—UEFISCDI, grant number PN-III-P4-ID-PCE-2020-0008.

**Conflicts of Interest:** The authors declare no conflict of interest.

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
