# Peer review of "Study on the Design Stage from a Dimensional and Energetic Point of View for a Marine Technical Water Generator Suitable for a Medium Size Container Ship"

_inventions, doi:10.3390/inventions8010022_

Round 1

Reviewer 1 Report

I have read through your MS and have come away wondering what you were trying to say and why I have a number of suggestions for improvement and some specific points

Specific Points

Line 29 is incorrect most seawater has between 25 and 45 g Nacl/L, please correct or give a source reference for your low value

Line 30-34. The principal commercial units are thermal, reverse osmosis , forward osmosis and freeze desalination approaches. The approaches like ultra filtration and electrocoagulation have been trialled. Commercial units using ZVI desalination (zero valent iron) and CO2 desalination are currently under trial (These are chemical approaches which do not require external energy and produce no reject brine). You should really address all these - suggest you try MDPI journals Sustainability, Hydrology, and Water for 2020 - 2022 for current articles on these.

Line 40 - please define your usable application at this point

Line 42 - mot sure about your units do you mean 5 ppm or do you mean 500 ppm. Your use of a comma is confusing, either use a full stop if you mean 5, or remove the comma if you mean 500. The same comment applies throughout the MS. Also you should standardise the units throughout the MS. I suggest you replace mg/L and ppm with g/L.

Suggestions for improvement

1. The Abstract needs to be rewritten. Please identify what you are intending to do with the MS and define the results of your study. As written it does not describe the content of your MS

2. Title - Please redefine. A technical water generator could mean anything

I assumed that you meant something like

Omocea, I., 2010. Exergetic Study of Naval Technical Water Generator. Journal of Marine Technology and Environment3(2).

and were describing a hot water heat recovery generator

Baldi, F.; Ahlgren, F.; Nguyen, T.-V.; Thern, M.; Andersson, K. Energy and Exergy Analysis of a Cruise Ship. Energies 201811, 2508. https://doi.org/10.3390/en11102508

However, I am not sure having read your MS that this is the case.

3. Introduction. I think that you should 1. summarise the problem you are trying to solve, 2. summarise how the issue has been addressed to date on ships. 3. clearly summarise the invention you are going to describe. - I suggest you rewrite the introduction using a typical patent pre-amble to the normal patent list of Figures and their description. 4. You should state the patent status of your invention and the application number or awarded number and priority date.

4. This paper is categorised as a research article, therefore you need to have a methodology. Section. This is absent from your MS

5. Summarise your invention - I presume that it is Figure 3.or is t Fig 2 or is it both?

You do not need to describe any other desalination process other that your invention in the MS therefore sections 2.2.3/4 should be deleted..

Section 3 should be rewitten, please use it to succinctly describe your invention. I presume fig is a summary of the invention placed in a ship. Your description is inadequate. Each piece of apparatus and flow line should be separately numbered and described using a standard patent text format. If you unclear, please read some patents on Google patents. This will give you the required style. This will allow lines 235 - 257 to be meaningful also 382 - 412

Line 362 seems to indicate that you are promoting a specific company's invention. The reader will now think that you are not describing an invention of your own or that you are trying to advertise a commercialised version of your invention. I am at a bit of a loss to understand what you are trying to achieve at this point.

In my view the MS is neither concise or focussed. It is unclear whether it is a review or an invention describing article.  It is unclear what the invention is. Figure 8 appears to indicate it is Aqua blue. The benefits are not clear. It appears to relate to electricity consumption. Again the text is not focussed, concise or clear. Please when you revise the MS put a reference to support each point made and each equation which is not your own.

Author Response

ANSWERS TO THE REVIEWERS' COMMENTS

Manuscript ID: inventions-2079371

Title: Study on the Design Stage from a Dimensional and Energetic Point of View for a Marine Technical Water Generator Suitable for a Medium Size Container Ship (retitled, as suggested)

GENERAL COMMENTS

A major revision has been carried out following carefully all the indications, suggestions and observations formulated by the reviewers.

The main changes operated are outlined using the yellow colour in the modified and keeping in mind the following suggestions, listed next:

1) As suggested, an overall revision of the text was carried out in order to correct some small mistakes all along the work. A spell check and editing of the English language and style was also carefully performed. Furthermore, many phrases and paragraphs have been reformulated in order to provide a better image of the objectives and of the results coming from the proposed work.

2) The abstract was entirely redesigned.

3) The Introduction has been completely redesigned. In the first instance the text didn’t had an well defined introduction, being considered by me the general description of sources of technical water onboard ships, just being listed nad breifly described.

4) Please allow me to point out that I tried to redefine the technical purpose of the invention presented in the study. In essence, the study consists in the dimensioning of a technical water generator. Using the established calculation principles, I tried to identify an optimal solution for a medium-sized container ship. In the later part of the studie I wanted to demonstrate that the identified technical solution can be optimized so as to increase the operation parameters of a main engine on board the ship, resumin my study just on the operating principles of any technical water generatpr.

Furthermore, it has to be also highlighted that the authors tried to follow all the suggestions and observations formulated by the reviewers and to operate (as much as it was possible) all the corrections indicated by them. In order to follow the corrections indicated and operated in the study, a version of the manuscript having all the changes operated tracked (highlighting them using the yellow Text Hilight Color in Microsoft Office Word) has been also uploaded together with the last form of the manuscript (without the changes highlighted in yellow color).

The specific corrections operated according to the suggestions of the reviewers are given next together with detailed explanations.

Reviewer 1

  1. Line 29 is incorrect most seawater has between 25 and 45 g Nacl/L, please correct or give a source reference for your low value.

Thank you for this observation. I analyzed your indications and, after a correct documentation, I replaced the value with the generally valid and accepted one in the specialized literature, which is in the range of 33 - 37 mg/l.

  1. Line 30-34. The principal commercial units are thermal, reverse osmosis, forward osmosis and freeze desalination approaches. The approaches like ultra filtration and electrocoagulation have been trialled. Commercial units using ZVI desalination (zero valent iron) and CO2 desalination are currently under trial (These are chemical approaches which do not require external energy and produce no reject brine). You should really address all these - suggest you try MDPI journals Sustainability, Hydrology, and Water for 2020 - 2022 for current articles on these

Once again, thank you for this observation. My basic idea was to simply list and briefly describe the main methods that can be used to obtain or generate technical water. Of course, not all of these technological processes can be applied on board a ship, given the limitations regarding the available space, the energy generated on board and other relevant aspects. However, I just wanted to point out that there are several principles and technologies that can be used to generate technical or even drinking water.

  1. Line 40 - please define your usable application at this point

Upon your observation, for which I will remain grateful, I made the indicated correction in line 40, summarizing only those methods and principles of obtaining technical water that are applicable on board a ship.

  1. The Abstract needs to be rewritten. Please identify what you are intending to do with the MS and define the results of your study. As written it does not describe the content of your MS

I have completely rewritten the abstract, according to your instructions, pointing out the importance of identifying optimal solutions for generating technical water on board the ship and the role of this installation in the operation of the main engine.

  1. Title - Please redefine. A technical water generator could mean anything

Once more, thank you for the observation. As you suggested, I considered the appropriate title for this study to be Study on the Design Stage from a Dimensional and Energetic Point of View for a Marine Technical Water Generator Suitable for a Medium Size Container Ship.

  1. Introduction. I think that you should 1. summarise the problem you are trying to solve, 2. summarise how the issue has been addressed to date on ships. 3. clearly summarise the invention you are going to describe. - I suggest you rewrite the introduction using a typical patent pre-amble to the normal patent list of Figures and their description. 4. You should state the patent status of your invention and the application number or awarded number and priority date

On your instructions I tried to summarize the central subject of the study, focusing only on the methods of obtaining technical water, which can be implemented on board the ship. At the same time, I also mentioned the state of technologies of this type, which have already been implemented in the maritime field. I reiterated the importance of the study in the form of a preamble and mentioned the status of this type of technical water generator. As it is a fairly common installation on board modern ships I do not hold any patents in this area.

  1. This paper is categorised as a research article, therefore you need to have a methodology. Section. This is absent from your MS

Again, thank you for this observation. Indeed, I omitted to specify the methodology for structuring the study. I tried to remedy the situation by formulating my own study structuring methodology.

  1. Summarise your invention - I presume that it is Figure 3.or is t Fig 2 or is it both?

Following your instructions, I opted for figure 4. Thus, I completely eliminated figures 2 and 3, thus figure 4 became figure 2, within the corrected subchapter. Figure 4 seemed more relevant to the context described in the following chapters.

  1. You do not need to describe any other desalination process other that your invention in the MS therefore sections 2.2.3/4 should be deleted..

Once more, thank you for the observation. I have deleted sub-sections 2.2.1 (Electrodialysis), 2.2.2 (Ultrafiltration), 2.2.3 (Desalination by freezing). Thus, the remaining sub-section, that became the main subject of this chapter is 2.2.4.

  1. Section 3 should be rewitten, please use it to succinctly describe your invention. I presume fig is a summary of the invention placed in a ship. Your description is inadequate. Each piece of apparatus and flow line should be separately numbered and described using a standard patent text format. If you unclear, please read some patents on Google patents. This will give you the required style. This will allow lines 235 - 257 to be meaningful also 382 - 412

Upon your observation, for which I which I thank you, I tried to adapt the content of this section. The system description has been systematized. I have not been able to adapt every element of Figure 5, which has now become Figure 3, as I believe it represents a generalized scheme that can be applied to many similar systems that already exist. The description and related calculation remained valid for the context of the analyzed system. At the same time, figure 6, which has now become figure 4, describes a simplistic system, similar to the one I wanted to achieve through the dimensioning calculation.

  1. Line 362 seems to indicate that you are promoting a specific company's invention. The reader will now think that you are not describing an invention of your own or that you are trying to advertise a commercialised version of your invention. I am at a bit of a loss to understand what you are trying to achieve at this point.

Indeed, the system described in line 362, which has now become line 351, is very similar to an existing system on the market, produced by the indicated company, but the reinterpretation tends towards the generalized system that I wanted to customize for the ship for which I have established the necesity. Through this calculation I tried to determine/calculate the technical characteristics of a technical water generation system that can be used on board the reference ship. I am not aiming for a specific generator, but, for establishing the technical characteristics, I was inspired by an already existing model, which proved its usefulness. In this manner I wanted to demonstrate that the calculation can be applied in the case of any other Panamax contaienr ship that needs such a system.

  1. In my view the MS is neither concise or focussed. It is unclear whether it is a review or an invention describing article.  It is unclear what the invention is. Figure 8 appears to indicate it is Aqua blue. The benefits are not clear. It appears to relate to electricity consumption. Again the text is not focussed, concise or clear. Please when you revise the MS put a reference to support each point made and each equation which is not your own.

As mentioned above, I tried to focus the study on the computational methodology (the calculation). It is inspired by specialized literature, but I tried to adapt it according to the studied ship or ships in this range (Panamax). In this regard, I tried to customize the calculation for this study. In order to eliminate any form of promotion of a type of technical water generator, I have given up all references to a specific type of technical water generator. I found it unnecessary to add additional references as I tried to customize the calculation as much as possible.

Reviewer 2 Report

This problem is relevant for the journal scope. The manuscript follows the formal regulations of the journal.

I suggest the major revision of the manuscript.

Remarks

1. Please emphasize the novelty side(s) of your manuscript.

2. I suggest the following the standard article structure: Introduction, Material and Methods, Results and Discussion, Conclusions.

3. I suggest putting over the calculations into the supplementary part.

4. Please avoid sketchy writing.

5. Please provide information on the accuracy of the results.

6. I suggest adding a Nomenclature part to the manuscript.

7. Please cite more papers from this journal in the last two years on a similar topic to this research.

Author Response

ANSWERS TO THE REVIEWERS' COMMENTS

Manuscript ID: inventions-2079371

Title: Study on the Design Stage from a Dimensional and Energetic Point of View for a Marine Technical Water Generator Suitable for a Medium Size Container Ship (retitled, as suggested)

GENERAL COMMENTS

A major revision has been carried out following carefully all the indications, suggestions and observations formulated by the reviewers.

The main changes operated are outlined using the yellow colour in the modified and keeping in mind the following suggestions, listed next:

1) As suggested, an overall revision of the text was carried out in order to correct some small mistakes all along the work. A spell check and editing of the English language and style was also carefully performed, while trying to avoid sketchy writing. Furthermore, many phrases and paragraphs have been reformulated in order to provide a better image of the objectives and of the results coming from the proposed work.

Furthermore, it has to be also highlighted that the author tried to follow all the suggestions and observations formulated by the reviewers and to operate (as much as it was possible) all the corrections indicated by them. In order to follow the corrections indicated and operated in the study, a version of the manuscript having all the changes operated tracked (highlighting them using the yellow Text Hilight Color in Microsoft Office Word) has been also uploaded together with the last form of the manuscript (without the changes highlighted in yellow color).

The specific corrections operated according to the suggestions of the reviewers are given next together with detailed explanations.

Reviewer 2

  1. Please emphasize the novelty side(s) of your manuscript

Thank you for this observation. In the abstract of the study I tried to point out the fact that the novelty of the study consists in the fact that it represents a customized calculation for a technical water generator for a certain type of medium-sized container ship. It can be valid for any container ship in this category, regardless of the type of technical water generator it has installed on board.

  1. I suggest the following the standard article structure: Introduction, Material and Methods, Results and Discussion, Conclusions

Once again, thank you for this observation. Using the structuring mode applied initially, in the first writing, I adapted the wording for the sub-chapter titles to correspond to the standard structure of such an article.

  1. I suggest putting over the calculations into the supplementary part

Upon your observation, for which I will remain grateful, I would like to point out that the calculation itself remains an important part of the article. This is indeed inspired by the specialized literature, but it is customized for my study on the medium-sized container ship case. For these reasons, I have dedicated a separate chapter to this calculation, being the one in which the specific results obtained are highlighted. In this regard, please allow me to devote a separate chapter to it, precisely with the idea of emphasizing its importance.

  1. Please avoid sketchy writing

In my effort of avoiding sketchy writing I have eliminated almost two pages form the article. Upon reviewing the style of writing I also noticed some word and phrases that were outside the intended context. To avoid any ambiguity I have eliminated entire phrases.

  1. Please provide information on the accuracy of the results

Once more, thank you for the observation. As you suggested, the calculation is an important part of this study. The accuracy of results is demonstrated by the specialized literature used that inspired it. On the other hand it as an adaption for the reference sized container ship and it’s accuracy allows me to predict the technical features of the technical water generator for the ship itself.

  1. Please cite more papers from this journal in the last two years on a similar topic to this research

Thank you for this observation. In the Reference section of this article I added four new titles of papers from this journal. These are titles which I found suitable for the subject of this article and have useful information.

Reviewer 3 Report

The manuscript is regarded as a brief technical report for onboard desalination techniques and processes, and by these means, it lays in the scope of the journal. My suggestion is that it can be published in Inventions in it's present form.  

Author Response

ANSWERS TO THE REVIEWERS' COMMENTS

Manuscript ID: inventions-2079371

Title: Study on the Design Stage from a Dimensional and Energetic Point of View for a Marine Technical Water Generator Suitable for a Medium Size Container Ship (retitled, as suggested)

GENERAL COMMENTS

A major revision has been carried out following carefully all the indications, suggestions and observations formulated by the reviewers No.1 and No.2.

Thank you for your observation. As you pointed out the manuscript is regarded as a brief technical report for onboard desalination techniques and processes, and by these means, it lays in the scope of the journal.

Thank you for your kind review.

Round 2

Reviewer 1 Report

General Comments

I have gone through your responses to my initial review. While the current MS is an improvement on its predecessor it is still not in a publishable form.

I found three points you made to the reviewer pertinent to your MS even though you did not articulate them in the MS. They are

Upon your observation, for which I which I thank you, I tried to adapt the content of this section. The system description has been systematized. I have not been able to adapt every element of Figure 5, which has now become Figure 3, as I believe it represents a generalized scheme that can be applied to many similar systems that already exist. The description and related calculation remained valid for the context of the analyzed system. At the same time, figure 6, which has now become figure 4, describes a simplistic system, similar to the one I wanted to achieve through the dimensioning calculation.

Indeed, the system described in line 362, which has now become line 351, is very similar to an existing system on the market, produced by the indicated company, but the reinterpretation tends towards the generalized system that I wanted to customize for the ship for which I have established the necesity. Through this calculation I tried to determine/calculate the technical characteristics of a technical water generation system that can be used on board the reference ship. I am not aiming for a specific generator, but, for establishing the technical characteristics, I was inspired by an already existing model, which proved its usefulness. In this manner I wanted to demonstrate that the calculation can be applied in the case of any other Panamax contaienr ship that needs such a system.

As mentioned above, I tried to focus the study on the computational methodology (the calculation). It is inspired by specialized literature, but I tried to adapt it according to the studied ship or ships in this range (Panamax). In this regard, I tried to customize the calculation for this study. In order to eliminate any form of promotion of a type of technical water generator, I have given up all references to a specific type of technical water generator. I found it unnecessary to add additional references as I tried to customize the calculation as much as possible

If you could condense these three paragraphs into a single focussed paragraph at the end of the introduction, this would I think allow the reader to see what you are intending to do in the MS and where you think the innovation you are intending to develop is.

In light of these paragraphs I think that on Line 1 you should replace the word "Article" with the term "Concept". The MS you have written falls within the MDPI definition of a conceptual paper.

In places your English is poor please ask a native English speaker to go through your text prior to resubmission.

I am not sure what the supplementary information file is for. It looks like another version of the main text.

The MS is still written at a very low technical level. The revised introduction would be expected to outline the nature of the problem you are trying to address. How it has been addressed by others (in patent with patent no. reference and academic references). Where you see a continuing issue and how you intend to address it. The introduction you provide does none of this.

At the end of the introduction (line 93) the reader is left wondering what the purpose of this study is. Please add a paragraph (as noted above) defining what you are intending to achieve with the study and why it is being undertaken.

In your comments to the reviewer you state

essence, the study consists in the dimensioning of a technical water generator. Using the established calculation principles, I tried to identify an optimal solution for a medium-sized container ship. In the later part of the studie I wanted to demonstrate that the identified technical solution can be optimized so as to increase the operation parameters of a main engine on board the ship, resumin my study just on the operating principles of any technical water generatpr.

This information should have been succinctly placed at the end of your introduction.

You describe the MS as a research article, yet to me it does not do this therefore your subheading 2.0 materials and methods is inappropriate. Change to 2.0 Study Structure

I did suggest in my earlier review that you should consult a series of patents in order to see how to write this paper. This does not appear to have been done.

None of your equations look as if they are your own. Please add a source reference for each equation used.

None of the figures are referred to in the text. Please adjust accordingly.

Figure 3 - please label all the flow lines etc in the figure and refer to each component in the text

Please check your units. Your salinity units throughout are expressed in mg/L when they should be in g/L. Your length and diameter units are in a mixture of imperial and metric. Please standardize using the metric version.

When writing a decimal you use the divider comma, this OK for papers written in French. Please replace with a full stop to bring the ms into standard scientific English, otherwise English, Australian and US readers will take it as a 1000 division, Indian readers will take it as a 100,000 division.

Please note that the revised text is full of typos and grammatical errors. Many of these will be caught if you use the MS spelling, grammar and accessibility functions.

Other comments

Line 23 typo engine is mispelt

Line 93 - please clearly define what you mean by a technical water generator within the introduction text. - this is to improve reader access. I am assuming that you mean a desalination plant.

Line 102 - global change -replace chapters with sections throughout the ms

Line 133 - most seawater is in the range 28 - 40 g/L. The range you provide of 33-37 mg/L is drinking quality water as defined by WHO, i.e. <400 mg/L.

Guidelines for drinking-water quality: Fourth edition incorporating the first and second addenda (who.int) 2022

Is that what you mean? It is excellent for product water but is about 1000 times lower than the salinity of seawater. I  think that you meant 33-37 g/L - please check

The Internationally accepted reference salinity for seawater is

Reference-Composition Salinity of exactly SR=35.16504 g kg−1. i.e. 35165.04 mg/L and not as you suggest 33-37 mg/L - see the attached reference

Millero, F.J., Feistel, R., Wright, D.G. and McDougall, T.J., 2008. The composition of Standard Seawater and the definition of the Reference-Composition Salinity Scale. Deep Sea Research Part I: Oceanographic Research Papers55(1), pp.50-72.

I raised this issue in my original review. You have incorrectly stated in response

Thank you for this observation. I analyzed your indications and, after a correct documentation, I replaced the value with the generally valid and accepted one in the specialized literature, which is in the range of 33 - 37 mg/l.

There is no specialised or reputable scientific literature which would state that the salinity of seawater is 33-37 mg/L.

Line 146 - should add the word water after technical

Line 147 - 149 - are you sure these numbers are correct. They seem to be very low and difficult to practically attain using the equipment described in the ms.

Line 159-163 - poor English - not sure that they are necessary. Either reword or delete.

My review copy of the MS jumps from line 165 to 193. Am I missing 30 lines?

Line 164 - 195 - poor English please reword to improve message clarity

Line 194 to 204 - I think that you could reduce this to a short sentence. There is probably about 60 - 70 million m3/d of desalinated water being produced in RO units. It is now a well established technology.

Line 204 to 230 could also be reduced to a single line remember that >100 m3 /d of potable municipal water are currently produced through desalination, most of it through the conversion of seawater. You are dealing with a mature technology on the whole, not a technology in its infancy.

Figure 1 is not referenced and appears out of place. Either delete or include an appropriate reference to it.

Line 232 -241 can be deleted. You are covering basic concepts very simplistically which should be common knowledge- Therefore this paragraph can be deleted.

Line 242 - should state Figure 2 illustrates the process flow for a RO unit. - please reword

Figure 2 caption - drawn by the authors can be removed

Line 255/6 remove second "in general"

Line 370 - Add "This is to support its 3 main roles"

Line 374 - 411 & Figure 3 - every element on the figure should be specifically identified with a number in the figure and referred to specifically in the text. Otherwise the reader will have difficulty understanding your figure and linking it to the text

Line 412 The operation of the three water installations onboard the medium sized container  should read The operation of each of the three water installations onboard the medium sized container

Line 422-428 - poor English can be deleted as it does not really add anything.

Line 433 - check spelling

Line 436 - you need a subheading indicating that you are doing an example calculation using the data in Table 1 together with a short sentence explaining what you are doing.

Line 436 - 454 this information should be transferred to a table. Please provide a reference to support each data value. Not sure that I agree with your values for q, q=0.7 L/S = 100 L/day per person. This normal for when people are on land and have washing machines, cooking, etc. On a ship many of these utilities are shared, therefore the net requirement should be less -please check and both clarify and justify your values in the revised text.

Line 470 - the standard units are metric if you want to use imperial (inches) please give the adjustment from metric to imperial., e.g. 1 inch = 2.54 cm.

Line 477 asperities is the wrong word. Please reword

Line 469, you are using the French approach of  using a comma to signify a change from a whole number to a fraction. In English scientific writing this should be a full stop 0,0729 becomes 0.0729. In English writing the use of a comma at this point signifies a size, which is 1000 times greater than your intent - please change throughout the ms.

Please put in a new heading around Line 428 - "Example Calculation" Please add an introductory paragraph to this section, and make sure that your example data set is placed in a table so the reader can easily see what information is required for your model.

In this section you have a number of equations. It would improve reader access if you placed the symbols and their unit in a table in the text at around this point. All tables should be referred to in the text.

Line 621 - You state

The technical water generator has an optimized technology that allows it to convert a higher percentage of technical water from salt water.

It is not clear to the reader at this point how this works and why this statement is true. However, this is by far the most important statement in your MS because it indicates you have achieved significant commercial novelty. Would it be possible to add a flow chart in the text or appendix indicating the conventional approach and then a similar flow chart (or pfd) illustrating the steps you took to achieve your innovation.

Line 626 replace diagram with Figure 6.

Figure 5 provides the Stype - please provide the relevant drawings for Ctype conventional plate and S&T and refer specifically in the Figure 6 caption to these figures.

In conclusion I think that you have a neat idea, but it is still very poorly explained and therefore difficult for another researcher to easily access what you are trying to say. I have made some suggestions as to how to improve the ms, and may have other comments once the revision has been concluded.

Author Response

ANSWERS TO THE REVIEWERS' COMMENTS (2)

Manuscript ID: inventions-2079371

Title: Study on the Design Stage from a Dimensional and Energetic Point of View for a Marine Technical Water Generator Suitable for a Medium Size Container Ship (retitled, as suggested)

GENERAL COMMENTS

First of all, thank you for your support throughout the process of writing this article/essay. Once again, we tried to take into account all your remarks. In this regard, we brought a series of major changes compared to the previously submitted version. Thus, another major revision has been carried out following carefully all the indications, suggestions and observations formulated by the reviewer.

The major changes operated can be summarized in the points specified below. They are also highlighted in a version of the new text.

1) The Introduction has been updated. I condensed the paragraphs pointed out in 2 phrases (a single focused paragraph). Following your remarkes I tried to find and study some patent examples. Google search engine helpt me out a little bit, but most of them were from Japan or Russia and this didn’t advantaged me. Overall, I hope I caught the main idea and tried to mentioned them as you pointed out.

2) English really does not give me an advantage, especially the technical language. I'm still trying to find my words. To improve the writing style, we turned to a more experienced colleague of ours, who helped us with a general proofreading after completing the text. I hope this has brought some improvement in the style of editing and expression..

3) Based on your instructions, we have made substantial changes to many of the paragraphs of the text. In the initial version that we sent, we highlighted and marked these changes in yellow. The indications pointed out by you involved a series of larger changes to some paragraphs, respectively less obvious changes to others, but I have marked them all.

The specific corrections operated according to the suggestions of the reviewers are given next together with detailed explanations.

  1. If you could condense these three paragraphs into a single focussed paragraph at the end of the introduction, this would I think allow the reader to see what you are intending to do in the MS and where you think the innovation you are intending to develop is.

Once again, thank you for this remark. The three paragraphs are now condensed in a shorter manner in the Introduction of this text.

  1. In places your English is poor please ask a native English speaker to go through your text prior to resubmission

I can't say otherwise. I really need to improve my English, but I tried, with the help of a colleague, to give coherence to the text and to adapt it in a technical manner. I don't think I succeeded in total, but I hope I managed to express the essential ideas in a more correct way.

  1. The MS is still written at a very low technical level. The revised introduction would be expected to outline the nature of the problem you are trying to address. How it has been addressed by others (in patent with patent no. reference and academic references). Where you see a continuing issue and how you intend to address it. The introduction you provide does none of this

Upon your observation, for which I’m grateful, I tried to correct this aspect. Following your overall instructions I think this has been improved. That is why, I studied some patents. I mentioned them in the Introduction and I also upgraded the References section by mentioning these patterns (their links) in the ending part.

  1. At the end of the introduction the reader is left wondering what the purpose of this study is. Please add a paragraph (as noted above) defining what you are intending to achieve with the study and why it is being undertaken.

Thank you for pointing this out for me. As I stated above, a big part of the Introduction has been modified. I hope I managed to express the purpose of this study in a more clearly manner, in order to ease the main ideas I intend to express.

  1. In your comments to the reviewer you state essence, the study consists in the dimensioning of a technical water generator. Using theestablished calculation principles, I tried to identify an optimal solution for a medium-sized container ship. In the later part of the studie I wanted to demonstrate that the identified technical solution can be optimized so as to increase the operation parameters of a main engine on board the ship, resumin my study just on the operating principles of any technical water generator. This information should have been succinctly placed at the end of your introduction.

Once more, thank you for the observation. As you suggested, I also condensed this entire statement/paragraph to a single line.

  1. You describe the MS as a research article, yet to me it does not do this therefore your subheading 2.0 materials and methods is inappropriate. Change to 2.0 Study Structure

Thank you again. The Materials and Methods subheading got changed to Study Structure

  1. I did suggest in my earlier review that you should consult a series of patents in order to see how to write this paper. This does not appear to have been done.

Again, thank you for this observation. We made an effort to find a series of patents developed in this field. We tried to extract the main ideas from them in order to make a short comparison with the previous attempts that were carried out in the field. The studied patents are mentioned in the introduction in a succinct version to emphasize how they resemble what we are trying to present.

  1. None of your equations look as if they are your own. Please add a source reference for each equation used

Upon your observation, for which I’m grateful, I tried to correct this aspect. The calculation method has been obtained by combining two methods exemplified in two Romanian books. There are now mentioned as No. 6 and N. 7 references.

  1. None of the figures are referred to in the text. Please adjust accordingly.

Once more, thank you for the observation. The figures have been referred to in the text before their apparence.

  1. Figure 3 - please label all the flow lines etc in the figure and refer to each component in the text

Upon your observation, for which I which I thank you, I marked all flow lines with small arrows and I numbered all components.

  1. Please check your units. Your salinity units throughout are expressed in mg/L when they should be in g/L. Your length and diameter units are in a mixture of imperial and metric. Please standardize using the metric version.

As you pointed out I corrected al salinity units.

  1. When writing a decimal you use the divider comma, this OK for papers written in French. Please replace with a full stop to bring the ms into standard scientific English, otherwise English, Australian and US readers will take it as a 1000 division, Indian readers will take it as a 100,000 division.

As mentioned above, I hope I’m not confusing the usage of coma in this context. The modification has been made. I’ve eliminated all comas and used only “.” for decimals.

  1. Please note that the revised text is full of typos and grammatical errors. Many of these will be caught if you use the MS spelling, grammar and accessibility functions.

Upon your observation, for which I’m grateful, I used the options of Microsof Office Word, in order to eliminate and re-formulate all misspelled word and expressions. This helpt me out, as I saw many typos and I hope I have eliminated them in the proper manner.

  1. Other comments:

- Line 23 typo engine is mispelt – The word has been revised;

- Line 93 - please clearly define what you mean by a technical water generator within the introduction text. - this is to improve reader access. I am assuming that you mean a desalination plant – As I mentioned, the Introduction has been modified, including this aspect;

- Line 102 - global change -replace chapters with sections throughout the ms – The replacement has been made;

- Line 133 - most seawater is in the range 28 - 40 g/L. The range you provide of 33-37 mg/L is drinking quality water as defined by WHO, i.e. <400 mg/L (Guidelines for drinking-water quality: Fourth edition incorporating the first and second addenda (who.int) 2022. Is that what you mean? It is excellent for product water but is about 1000 times lower than the salinity of seawater. I  think that you meant 33-37 g/L - please check. The Internationally accepted reference salinity for seawater is Reference-Composition Salinity of exactly SR=35.16504 g kg−1. i.e. 35165.04 mg/L and not as you suggest 33-37 mg/L - see the attached reference Millero, F.J., Feistel, R., Wright, D.G. and McDougall, T.J., 2008. The composition of Standard Seawater and the definition of the Reference-Composition Salinity Scale. Deep Sea Research Part I: Oceanographic Research Papers55(1), pp.50-72. I raised this issue in my original review. You have incorrectly stated in response. Thank you for this observation. I analyzed your indications and, after a correct documentation, I replaced the value with the generally valid and accepted one in the specialized literature, which is in the range of 33 - 37 mg/l. There is no specialised or reputable scientific literature which would state that the salinity of seawater is 33-37 mg/L – This is a very usefull remark, that is why I thank you for it and, yes: you mentioned it before. As you also mentioned, I opted for a standard interval that I found in the specialized literature, only that I made a major confusion between the units of measure in the initial part of the project and continued with it throughout the whole text. The correct variant is the one indicated by you, which I have also used according to the indications you expressed now.

- Line 146 - should add the word water after technical – The missing word was added.

- Line 147 - 149 - are you sure these numbers are correct. They seem to be very low and difficult to practically attain using the equipment described in the ms - Thank you for pointing this out. Even if the values do not seem optimal for this type of calculation, the basic idea that I want to express to the reader is the calculation methodology. However, from the experience I had with this type of system, I could confirm that they are acceptable values.

- Line 159-163 - poor English - not sure that they are necessary. Either reword or delete – This paragraph has been re-written and some part deleted.

- My review copy of the MS jumps from line 165 to 193. Am I missing 30 lines? – I’m sorry if I misled you, but in my version all seems to be numbered in a corect order. It is possible that your line counters have shifted, a fact that misled me when I was trying to identify the lines you refer to from this point on. I hope I have identified the lines you pointed out in a similar manner, but it is possible that I have confused some of them, especially if you pointed out lin 273 in your text version and for me 273 was in fact 289 (for an example). If this is the case, please reformulate the modification that I had to take care of, but I mistook them for others.

- Line 164 - 195 - poor English please reword to improve message clarity – Thank for your observation. The entire paragraph has been revised.

- Line 194 to 204 - I think that you could reduce this to a short sentence. There is probably about 60 - 70 million m3/d of desalinated water being produced in RO units. It is now a well established technology – As pointed out this was reduced in a meaningfull manner.

- Line 204 to 230 could also be reduced to a single line remember that >100 m3 /d of potable municipal water are currently produced through desalination, most of it through the conversion of seawater. You are dealing with a mature technology on the whole, not a technology in its infancy – As mentioned above, this was also been reduced in the same way.

- Figure 1 is not referenced and appears out of place. Either delete or include an appropriate reference to it – A reference to this figure has been made in the text before it.

- Line 232 - 241 can be deleted. You are covering basic concepts very simplistically which should be common knowledge- Therefore this paragraph can be deleted – The entire 9 lines have been deleted.

- Line 242 - should state Figure 2 illustrates the process flow for a RO unit. - please reword – Thank you for your suggestion. The statement has been made.

- Figure 2 caption - drawn by the authors can be removed – This has been removed and the reference has been indicated.

- Line 255/6 remove second "in general" – The corection has been made.

- Line 370 - Add "This is to support its 3 main roles" – The sugested text has been added;

- Line 374 - 411 & Figure 3 - every element on the figure should be specifically identified with a number in the figure and referred to specifically in the text. Otherwise the reader will have difficulty understanding your figure and linking it to the text – Thank you for this important sugestion. All elements have a certain number on their behalf. These numbers have also been market in the diagram;

- Line 412 The operation of the three water installations onboard the medium sized container  should read The operation of each of the three water installations onboard the medium sized container – The modification has been made as pointed out;

- Line 422-428 - poor English can be deleted as it does not really add anything – Deleted;

- Line 433 - check spelling – I have done a re-check on the spelling. As pointed out, some adjustements were needed;

- Line 436 - you need a subheading indicating that you are doing an example calculation using the data in Table 1 together with a short sentence explaining what you are doing – I’m very sorry and please excuse me. I don’t know if I really understood this modification. I had difficulties in trying to identify Table 1. That is why I couldn’t carry it out.

- Line 436 - 454 this information should be transferred to a table. Please provide a reference to support each data value. Not sure that I agree with your values for q, q=0.7 L/S = 100 L/day per person. This normal for when people are on land and have washing machines, cooking, etc. On a ship many of these utilities are shared, therefore the net requirement should be less -please check and both clarify and justify your values in the revised text – Once again, please excuse me, but I haven’t managed to identify the information. Please indicate to me again the lines where I can find the information where the data is found;

- Line 470 - the standard units are metric if you want to use imperial (inches) please give the adjustment from metric to imperial., e.g. 1 inch = 2.54 cm – All unit conversions have been made form imperial to metric;

- Line 477 asperities is the wrong word. Please reword – Thank you for pointing this mistranslation from Romanian. The word I meant for was “roughness” and I made the replacement;

- Line 469, you are using the French approach of  using a comma to signify a change from a whole number to a fraction. In English scientific writing this should be a full stop 0,0729 becomes 0.0729. In English writing the use of a comma at this point signifies a size, which is 1000 times greater than your intent - please change throughout the ms – All comas have been eliminated and I hope I haven’t missed anything. The Romanian system is similar to the French one and I haven’t knew about this until you told me this. Thanks you and I’ll have it in mind for future similar article;

- Please put in a new heading around Line 428 - "Example Calculation" Please add an introductory paragraph to this section, and make sure that your example data set is placed in a table so the reader can easily see what information is required for your model – Once again, I apologize for it, but I think our line numbering has does not fit. I have difficulties in identiying the Line 428 you wanted to make the adjustment on;

- Line 621 - You state - The technical water generator has an optimized technology that allows it to convert a higher percentage of technical water from salt water. It is not clear to the reader at this point how this works and why this statement is true. However, this is by far the most important statement in your MS because it indicates you have achieved significant commercial novelty. Would it be possible to add a flow chart in the text or appendix indicating the conventional approach and then a similar flow chart (or pfd) illustrating the steps you took to achieve your innovation – I’m afraid I haven’t really understood this recommendation. In order to develop this flowchart I have to comprehend the conventional approach. I will try to develop this chart, I would need a little guidance from you. Thank you in advance;

- Line 626 replace diagram with Figure 6 – The word diagram has been replaced with Figure;

- Figure 5 provides the Stype - please provide the relevant drawings for Ctype conventional plate and S&T and refer specifically in the Figure 6 caption to these figures – Thank you for this remark, but, to be honest with you, I have difficulties in finding this information. I checked all information and brochures from the manunfacturer, but it seems this information is not available, or I’m not looking in the right places. If you can spare the time, please share this information with me.

Reviewer 2 Report

The authors addressed properly the comments and the paper can be considered for publication.

Author Response

Thank you very much for your response and comments.

Round 3

Reviewer 1 Report

I hope you have a happy and prosperous New Year.

I have gone through your revised MS.

The MS is an improvement on the earlier version.

Introduction - there are typos on line 59, 63, 81,223 On line 62 the word profile should be omitted.

Line 80 replace "came to live" with "was used"

Line 156/157 - I still think your values look very low. I think that most boiler feed water has a higher salinity in the order of 150 - 600 mg/L . The US Navy has operated with a guide value of 25 grains per gallon (427 mg/L). Therefore you need to give reference support for each of your numbers or change them. There is extensive literature on this and the ability of 316 steel, etc to withstand salinity. Google scholar will give you access to suitable references. If you can achieved 5 ppm as per Line 630, please insert in line 158 before desalination the following. The conceptual process described in this study is designed to produce boiler feed water with a salinity of <5 ppm.

line 236 refer to Figure 2 not the diagram

Line 238 - typo

Line 242 replace diagram with Figure 2

Line 254 installe should be installed

line 356 - 372; 436-441 - Please give a source reference for each selected value which is not your own. If the numbers are your own please give a rationale for their selection so that others can adopt your approach. This data could be placed in a table

Line 481 replace chapter with section

Line 552 replace points out with demonstrates

Figure 6 - You need to provide a drawing similar to Figure 5 of the Aqua Blue c type so that the reader can easily compare the two and visualise your improvement as defined in Figure 7

The MS as it stands is a major improvement on the original. Your English style is not concise or focussed, and in many places is at a low technical level (i.e. at the level of reinventing the wheel). This does detract from the message you are putting forward. I have put forward some suggestions for improvement, but have largely ignored changes to your writing style in this review.

Author Response

ANSWERS TO THE REVIEWERS' COMMENTS (3)

Manuscript ID: inventions-2079371

Title: Study on the Design Stage from a Dimensional and Energetic Point of View for a Marine Technical Water Generator Suitable for a Medium Size Container Ship (retitled, as suggested)

GENERAL COMMENTS

Thank you once again for your understanding and patience, which you showed by the necessary indications for the correction of our essay/article. Considering the fact that the indicated changes were not major, as you also pointed out, we no longer highlighted the changes we made in yellow. The following changes were made to the text:

  1. Line 156/157 - I still think your values look very low. I think that most boiler feed water has a higher salinity in the order of 150 - 600 mg/L . The US Navy has operated with a guide value of 25 grains per gallon (427 mg/L). Therefore you need to give reference support for each of your numbers or change them. There is extensive literature on this and the ability of 316 steel, etc to withstand salinity. Google scholar will give you access to suitable references. If you can achieved 5 ppm as per Line 630, please insert in line 158 before desalination the following. The conceptual process described in this study is designed to produce boiler feed water with a salinity of <5 ppm.

Thank you for this remark. We followed your instructions and entered the mentioned expression. We again studied the references used for the values I established. Indeed, you are right, but from the sources we studied, the reference values vary from case to case. In the first version, we tried to establish a somewhat average value, but its correctness can be challenged.

  1. Line 356 - 372; 436-441 - Please give a source reference for each selected value which is not your own. If the numbers are your own please give a rationale for their selection so that others can adopt your approach. This data could be placed in a table

Upon your observation, for which I’m grateful, we pointed out the values are established according to the indicated bibliographic sources. It’s an marine installations/system published the Maritime University from ConstanÈ›a.

  1. Figure 6 - You need to provide a drawing similar to Figure 5 of the Aqua Blue c type so that the reader can easily compare the two and visualise your improvement as defined in Figure 7

Thank you for pointing this out for us. We found a drawing as pointed out, form reference no. [39].

  1. Other comments:

- Introduction - there are typos on line 59, 63, 81,223 On line 62 the word profile should be omitted -  typos corrected as you pointed out. Than you for indicating them to us;

- Line 80 replace "came to live" with "was used" – phrase replaced;

- Line 102 - global change - replace chapters with sections throughout the ms – The replacement has been made – phrase replaced;

- Line 236 refer to Figure 2 not the diagram – referenced as pointed out;

- Line 242 replace diagram with Figure 2 – word “diagram” replaced as indicated;

- Line 254 installe should be installed – typo corrected;

- Line 481 replace chapter with section – replaced as pointed out;

- Line 552 replace points out with demonstrates – replaced at you advice.
